# Spray-Dried, Nanoencapsulated, Multi-Drug Anti-Tuberculosis Therapy Aimed at Once Weekly Administration for the Duration of Treatment

**DOI:** 10.3390/nano9081167

**Published:** 2019-08-15

**Authors:** Lonji Kalombo, Yolandy Lemmer, Boitumelo Semete-Makokotlela, Bathabile Ramalapa, Patric Nkuna, Laetitia L.L.I.J. Booysen, Saloshnee Naidoo, Rose Hayeshi, Jan A. Verschoor, Hulda S. Swai

**Affiliations:** 1Council for Scientific and Industrial Research, Pretoria 0002, South Africa; 2Department of Biochemistry, Genetics and Microbiology, University of Pretoria, Pretoria 0002, South Africa

**Keywords:** nanomedicine, tuberculosis, spray-drying technology, efficacy, dose frequency

## Abstract

Aiming to improve the treatment outcomes of current daily tuberculosis (TB) chemotherapy over several months, we investigated whether nanoencapsulation of existing drugs would allow decreasing the treatment frequency to weekly, thereby ultimately improving patient compliance. Nanoencapsulation of three first-line anti-TB drugs was achieved by a unique, scalable spray-drying technology forming free-flowing powders in the nanometer range with encapsulation efficiencies of 82, 75, and 62% respectively for rifampicin, pyrazinamide, and isoniazid. In a pre-clinical study on TB infected mice, we demonstrate that the encapsulated drugs, administered once weekly for nine weeks, showed comparable efficacy to daily treatment with free drugs over the same experimental period. Both treatment approaches had equivalent outcomes for resolution of inflammation associated with the infection of lungs and spleens. These results demonstrate how scalable technology could be used to manufacture nanoencapsulated drugs. The formulations may be used to reduce the oral dose frequency from daily to once weekly in order to treat uncomplicated TB.

## 1. Introduction

Tuberculosis (TB) is arguably the best established human infectious disease in existence. The disease, caused by infection, usually of the lungs, with the bacterium *M. tuberculosis (M.tb)*, has followed humanity for millennia [1]. The World Health Organisation (WHO) estimated that 10 million people fell ill and ~1.6 million died from tuberculosis in 2017. Despite the availability of treatment and a partially effective vaccine, TB is still the leading cause of death from a single infectious agent globally [2].

The current drug treatment regimens are effective for drug-susceptible TB, but its duration over months and toxic side effects increase the risk of non-compliance with concomitant relapse and development of drug resistance [3]. The increasing occurrence of drug-resistant TB is of great concern in endemic areas, in particular to people who are more exposed, such as clinical staff or mine workers, and those who are immune compromised. The success rate in drug-resistant TB treatment is compromised by the challenge of the lesser bactericidal effect of the second- and third-line drugs combined with their more adverse systemic side effects [4].

In order to reduce the TB deaths by 90% by 2030 as promoted by WHO [2], an increased effort of the development of novel drugs for TB therapy or reformulation of existing drugs is needed. Reformulation of chemotherapeutics into novel pharmaceutical systems has enjoyed much attention over the past few decades. The opportunity in such systems to address limitations of the molecules, such as low bioavailability, short half-life, low efficacy and hepatotoxicity can add new efficacy and utility to existing drugs. Numerous anti-TB drug delivery nanosystems have been developed for this purpose with confirmed in vitro and in vivo successes [5,6,7]. Various approaches have been described that include pulmonary, oral, parenteral, and topical routes of administration. A recent review by Marinda et al. indicated the potential that pulmonary drug delivery could offer in the field of anti-TB therapy [8]. Even though every route has its advantages and limitations, the oral route remains the most cost-effective and acceptable by patients [9,10]. For oral delivery, the diverse physiology of the intestines should be considered when designing a nanoparticle for drug encapsulation. Factors such as variation in pH, varying mucus thickness, and structure, as well as numerous cell types, play a role whether these particles will be taken up into the system [11].

One recent success of an orally delivered nanoencapsulated drug system was demonstrated by Horvati and colleagues (2015) who reported a Poly DL-lactic-co-glycolic acid polymer (PLGA) nanoparticle for the encapsulation of a pyridopyrimidine derivative showing improved anti-mycobacterial activity and low toxicity compared to the unencapsulated drug in an infected guinea pig model [12]. Even a PLGA encapsulated anti-mycobacterial analogue of isoniazid (INH) showed an improved minimum inhibitory concentration (MIC) outcome, which the authors alluded to the drug-loaded nanoparticles being able to target intra- and -extracellular mycobacteria [13]. Different nanosystems have been developed for anti-TB drug delivery. Alongside PLGA, various other encapsulation nanosystems and approaches have been used to improve the outcome of the anti-TB drugs (ATDs). Examples such as liposomes/niosomes, polymeric nanoparticles, microspheres composed of different polymers (PLGA, alginate, gelatin and SLNs) have been extensively described in various reviews such as reportedby Costa et al. 2016 [3]. For instance, the use of flower-like polymeric micelles to improve the oral bioavailability of rifampicin was achieved by improving the water solubility and stability of the drug at acidic conditions showing a 3-fold improvement when administered with INH, compared to the free rifampicin suspension with the INH [14].

The entrapment of the therapeutic agents in biodegradable polymeric nano-systems designed to enhance targeted entry, slow-release, and retention of the antibiotics in the cells for longer periods should not only achieve an improved minimum inhibitory concentration, but also reduce dose frequency. Thus, it should improve patient compliance and reduce systemic side effects associated with conventional free anti-TB drugs. Previously, we have demonstrated the safety of orally administered uncoated PLGA nanoparticles in an in vivo mouse model. No pathological lesions or inflammation as assessed by histopathology assays were observed, compared to controls. It was also illustrated in the same study how the migration of these particles occurred from the gastrointestinal tract after oral administration to various organs of the body [15]. When PEG-coated PLGA nanoparticles were administered via the oral and intra-peritoneal route to unchallenged mice, the cytokine release profiles indicated induction of negligible inflammatory activity, thereby qualifying these PLGA particles as safe nanocarriers for anti-TB drugs [16]. After incorporating isoniazid or rifampicin into the PLGA nanoparticles, oral administration to mice showed a sustained drug release profile over a period of five days above the MIC, where the free drugs decreased to below MIC within 16 h [17,18]. In the present study, we assess the efficacy of the encapsulated first-line anti-TB drugs produced by means of a novel spray-dried nano-formulation in a weekly dose compared to the daily conventional dosages in a TB-infected mouse model. The spray-drying technology was selected owing to its simplicity, fewer process steps as well as shorter time to reach the desired state of a free-flowing powder as depicted in the diagram in Appendix A
Figure A1, unlike the freeze-drying technology that requires more than 72 h to get a dried powder.

## 2. Materials and Methods

### 2.1. Chemicals and Reagents

Rifampicin (RIF), Pyrazinamide (PZA), Isoniazid (INH), PLGA 50:50 (M_W_: 45,000–75,000), partially hydrolysed (87–89%) Polyvinyl alcohol (PVA) MW 13,000–23,000, polyethylene glycol (PEG) M_W_ 9 000, lactose monohydrate, Chitosan (85% de-acetylated, low viscosity), glycerol, Tween 20, Pluronic F127™ and salts for the preparation of phosphate buffered saline were purchased from Sigma-Aldrich Chemical Co., (St Louis, MO, USA). Difco Middlebrook 7H9 medium and OADC (Oleic Albumin Dextrose Catalase) were from Becton Dickinson and Company (Le Pont De Claix, France).

### 2.2. Nanoparticle Preparation and Characterisation

For the preparation of first-line anti-TB drugs encapsulated into nanoparticles, a modified double emulsion solvent evaporation spray-drying technique as described in Semete et al. (2012) was used [17]. Briefly, nanoparticles were prepared by first dissolving 100 mg of PLGA 50:50 (MW: 45,000–75,000) into 8 mL of ethyl acetate. To form the first emulsion, 2 mL of phosphate buffered saline (pH 7.4) was added and emulsified with a high-speed homogeniser with varying speed between 3000 and 5000 rpm. To the resultant water-in-oil (w/o) emulsion 1% w/v polyvinyl alcohol (PVA) was added and further emulsified at 8000 rpm. The double emulsion (w/o/w) was spray-dried at 95–110 °C with an atomising pressure between 5 and 8 bar. For the addition of the drugs, micelles containing actives were prepared and added to the first emulsion. To form micelles for pre-encapsulating hydrophilic drugs (INH or PZA), a few drops of Pluronic F127 were added. For the hydrophobic drug RIF, this was achieved by the addition of a few drops of Tween 20. For polyethylene glycol (PEG) coating, a 40 mL mixture was prepared consisting of 5 mL of 1% w/v PEG, 10 mL of 5% w/v lactose, 15 mL of 1% w/v PVA and 10 mL of 0.3% w/v chitosan during the second emulsion step. The first w/o emulsion was then dispersed in this mixture and emulsified. For the addition of the drugs, micelles containing actives were prepared and added to the first emulsion. The second w/o/w emulsion was obtained and immediately spray dried. Subsequent nanoparticles obtained from the spray drying process were characterised for size and size distribution by making use of dynamic light scattering by means of a Zetasizer Nano ZS instrument (Malvern Instruments Ltd., Malvern, Worcestershire, UK). In addition, surface charge measurements were determined by means of laser Doppler electrophoresis. The surface topography of nanoparticles was observed by means of a Digital Instruments Multimode NanoScope Version IV atomic force microscope (AFM) using phosphorous (n) doped Si cantilever probes. The tests were performed in a contact mode after a droplet of the nanoparticle suspension in distilled water was allowed to evaporate. The encapsulation efficiency and drug loading were determined as previously described [18].

### 2.3. In Vivo Assays

#### 2.3.1. Mice

C57Bl/6 female mice were bred under specific pathogen-free conditions at the University of Cape Town and used between 8–12 weeks. Infected animals were maintained in individually ventilated cages under biosafety level 3 conditions. All experiments were approved by the Research Ethics Committee of the University of Cape Town, Cape Town, South Africa.

#### 2.3.2. Mycobacteria and Infection

*M.tb* H37Rv was grown in Difco Middlebrook 7H9 medium containing 0.5% glycerol and enriched with 10% OADC. Cultures were incubated at 37 °C and grown until log phase, aliquoted and maintained as frozen stocks at −70 °C. For infection, an aliquot of *M.tb* H37Rv was rapidly thawed at 37 °C, passed 30 times through a 29.5 G needle and diluted in sterile saline. Mice were infected using a Glas-Col Inhalation Exposure System Model A4224. The pulmonary infection dose of 50–100 cfu/lung was confirmed by 10 mice one day after infection and plating the homogenised lung tissue on Difco Middlebrook 7H10 agar plates in 10-fold serial dilutions. Plates were semi-sealed in plastic bags, incubated for 17–21 days at 37 °C, after which the number of mycobacterial colonies were counted, and the infection dose calculated.

#### 2.3.3. Chemotherapy Preparation and Treatment

Isoniazid containing PLGA (INH-PLGA), rifampicin containing PLGA (RIF-PLGA) and pyrazinamide containing PLGA (PZA-PLGA) nanoparticles were prepared in 20% v/v Tween/Saline solution at concentrations based on the daily standard therapy doses prescribed for human adults. Empty PLGA particles were similarly prepared. For concentration calculations, the encapsulated drug weight and not the total particle weight was used. Encapsulated nanoparticles were therefore prepared at the following concentrations: INH-PLGA = 5 mg/kg/day, RIF-PLGA = 10 mg/kg/day, PZA-PLGA = 20 mg/kg/day.

For preparation of chemotherapy in drinking water, it was assumed that a 20 g mouse would drink 2.5 mL–5 mL water/day. Chemotherapy in drinking water was therefore prepared at the following concentrations: INH = 40 mg/L, RIF = 80 mg/L, and PZA = 160 mg/L to provide a final therapeutic dose of INH = 5–10 mg/kg, RIF = 10–20 mg/kg, and PZA = 20–40 mg/kg.

Upon infection with *M.tb* H37Rv at a dose of 50–100 cfu/lung, the infection was allowed to be established for three weeks and the bacilli burden determined. Thereafter, infected animals received therapy for durations of four weeks, six weeks, and nine weeks respectively in three separate experiments. Each experiment (n = 10) contained the following treatment groups namely (1) untreated (UT-control), (2) empty particles only (NP-control), (3) chemotherapy in water (CT-water), (4) encapsulated particles chemotherapy (CT-NP). Empty nanoparticles and encapsulated drug particles were administered in a volume of 200 µl by gavage every seven days to respective groups at doses 7x the daily dosage to ensure sufficient drug availability for the 7-day duration. CT-water was always available in the drinking water of the animals in that group for the treatment duration.

At the end of treatment, all animals were sacrificed by halothane administration. The spleens and lungs were removed and weighed. The left lung lobes were fixed in 4% paraformaldehyde for sectioning and subsequent hemoxylin/eosin and Ziehl Neelson staining for acid-fast bacilli. The rest of the lungs were processed for bacilli burden determination.

### 2.4. Statistical Analysis

The data are expressed as the mean ± SEM. Statistical analysis was performed by ANOVA (one-way analysis of variance, Tukey’s multiple comparison test) using GraphPad Prism software (version 4.01). For all tests, significance was determined at p-value < 0.05.

## 3. Results

### 3.1. Nanoparticle Formulations

Free-flowing powder of drug-loaded polymeric nanoparticles was obtained by a novel technique consisting of spray drying a multiple emulsion. It is extensively reported in literature that for a spray-drying process using a pneumatic or multi-fluid nozzle for atomisation of a liquid or suspension, the obtained droplet size is influenced by the feeding rate, surface tension and viscosity of the feed solution, the nozzle diameter as well as the pressure and density of the atomising fluid (air or inert gas such as nitrogen). Usually, the finest droplet size that can be achieved with the current nozzle configuration ranges between 10–100 μm. The average particle size D50 of a sample can be predicted by Lubanska’s equation as follows:(1)$$D_{50}=K_{d}D_{n}\;\sqrt{\frac{\eta_{f}}{\eta_{g}\times W_{e}}\left(1+\;\frac{F_{f}}{F_{g}}\right)}$$
where *K_d_* is a constant, *D_n_* is the orifice diameter of the nozzle, and *ŋ_f_* and *ŋ_g_* are kinematic viscosity of feed and gas, respectively. *F_f_* and *F_g_* are the flow rates of feed and gas whereas *W_e_* is the Weber dimensionless number correlating the gas pressure acting on the liquid to the liquid capillary pressure (surface tension) [19]. Several factors can be manipulated to reduce the droplet size, but the surface tension of the liquid that is being atomised is paramount. The droplet size is linked to the size of the derived dried particle through the following expression:(2)$$D_{p}=D_{d}\;\;\sqrt[{3}]{Y_{s}\left(\frac{\rho_{d}}{\rho_{p}}\right)}$$
here, *D_p_* is the diameter of the dried particle, *D_d_* is the diameter of the droplet before drying, *Y_s_* is the solid concentration in the spray solution, *ρ_d_* is the spray solution droplet density and *ρ_p_* the density of the dried particle [20]. In order to achieve nanoparticles in a free-flowing powder state, the spray solution should be atomised finely, which can only be achieved by starting with a stable emulsion.

All formulations yielded free-flowing powder after spray-drying. As depicted in Table 1, particle size for all the samples were below 500 nm with an acceptable polydispersity index between 0.2 and 0.4. A positive charge was obtained owing most likely to the presence of the natural polycation chitosan in the formulation. The range of zeta potentials observed lead to moderately stable nanosuspensions attributed to repulsive electrostatic forces exerted between the nanoparticles. The spray drying process resulted in an improved powder collection yield due to the design of the Buchi high performance cyclone. The relatively high encapsulation efficiency for RIF was attributed to its hydrophobic nature, which is similar to PLGA. These formulations were readily re-dispersed in aqueous dosage solutions resulting in translucent nanosuspensions that were easy to administer.

Evaluation of the surface morphology of the particles was conducted using atomic force microscopy (AFM).

All the formulations prepared for the study had a similar topography as presented by Figure 1. The scale at the top right corner of the image is the Z scale depicting the features that are protruding on the surface. The line scan of the image showed a relatively smooth surface on top of the particle. A small standard deviation was obtained illustrating an overall smooth morphology of the particles.

### 3.2. In Vivo Study

In this study, the bactericidal efficacy of INH-PLGA, RIF-PLGA, and PZA-PLGA nanoparticles against *M.tb* H37Rv in an aerosol inhalation challenge mouse model were investigated. After a 3-week infection period, all animals were treated for nine weeks thereafter. The treatment consisted of a daily oral dose of free INH/RIF/PZA (abbreviated CT-water) and a once-off weekly dose of the encapsulated drugs (abbreviated CT-NP). Untreated (UT) mice and empty nanoparticles (NP) served as controls. Each test group consisted of 10 mice.

The *M.tb* H37Rv infectious dose administered in all three experiments was verified to be between 50–100 cfu/lung by enumeration of colonies obtained after 18–24 h infection by serial dilution of homogenised lung tissue. After three weeks, the establishment of pulmonary infection was calculated to be between 1 × 10^5^–1 × 10^6^ cfu/lung and dissemination to the spleen averaged at 3 × 10^3^ cfu/spleen. These were normal and well within published ranges. The overall kinetic changes in lung, spleen and body weights over the experimental period are represented in Figure 2. Bodyweights measured for all groups over the total experimental period for any of the groups showed no significant differences with animal weight ranging from 20–26 g. All infected groups gained weight over the experimental duration (Figure 2C).

It is well established that infection will induce inflammation and recruitment of cells to sites of infection. Here lung and spleen weights were used as surrogate markers to reflect increased inflammatory responses as measured in Figure 2A,B.

Figure 2 shows that no significant differences were observed in spleen weights between any of the groups after four weeks of treatment, i.e., at week seven post-infection (*p* > 0.05). Measurement of lung weights reflected significant differences (*p*< 0.01) between UT-control groups and CT-water control groups indicating the resolution of inflammation in the CT-water treated group. Similarly, a significant difference (*p*< 0.01) between NP-control treated mice and CT-NP treated mice were found indicating the resolution of inflammation in CT-NP treated mice. There was no significant difference between the UT-control and NP-control treated mice indicating that the treatment with the empty particles did not influence the resolution of inflammation. Therefore, the particles in the lungs appeared to be inert and did not contribute to enhanced inflammation. It was noted that the lung weights in the CT-water treated group was significantly lower (*p*< 0.01) than that of the CT-NP treated group at this time point. This indicated that resolution of inflammation in the CT-water treated group happened faster than that of the CT-NP treated animals.

At six weeks after treatment i.e., nine weeks post-infection similar observations were noted with respect to lung and spleen weights as at four weeks, again with no significant differences in weights noted in the spleens of the different groups. However, the results differed in one respect, in that comparison of lung weights between the CT-water treated group and the CT-NP treated group no longer showed a significant difference after six weeks of treatment, indicating an equivalent resolution of inflammation at the longer treatment time point. These two groups also showed significant differences towards their control counterparts.

At nine weeks after treatment, i.e., 12 weeks post-infection, lung weights essentially reflected the same differences obtained at six weeks after treatment. However, for the first time a significant difference (*p*< 0.001) was noted in spleen weights between the UT-control group and CT-water group, indicating the resolution of inflammation in the spleen in the CT-water treated animals. Interestingly, significant spleen weight differences (*p*< 0.05) were also noted between the NP-control group and the CT-NP treated group, suggesting resolution of splenic inflammation in animals treated with drug encapsulated particles. Spleen weights of both the UT-water and NP-control groups were not significantly different, again indicating no enhancing effect by the empty nanoparticles on the induction of inflammation. Importantly, resolution of splenic inflammation in CT-water treated and CT-NP treated groups were found to be equivalent, indicating that both treatment approaches were effective for resolution of TB induced inflammation associated with the spleen after nine weeks of treatment.

*M.tb* bacterial burdens of the lungs and spleen were assessed in the different infected groups after four weeks (Figure 3A), six weeks (Figure 3B), and nine weeks (Figure 3C) of chemotherapeutic treatment. From the data at four weeks, it was noted that both the groups that received no chemotherapeutic treatment had equivalent bacterial lung and splenic bacilli burdens at 1 × 10^5^–1 × 10^6^ cfu/lung and 1 × 10^4^–1 × 10^5^ cfu/spleen respectively. A significant difference (*p*< 0.001) between the UT-control group and the CT-water group was noted with bacilli burdens 10–100 times lower in the CT-water treated group for both the lungs and spleens, indicating effective inhibition of bacterial replication when drugs were administered free in drinking water. Moreover, significant lower bacilli burdens were also noted in the CT-NP treated group compared to the NP-control group (*p*< 0.001) or the UT-control group (*p*< 0.01). Although the mean of the CT-NP treated group was higher than that of the CT-water treated group, there was no significant difference between the two groups for either the lungs or the spleens. Similar differences were noted after six weeks of chemotherapeutic treatment. The overall bacilli burden for the lungs was however 10–100 times lower after six weeks treatment compared to four weeks treatment in all the groups that received chemotherapy either as encapsulated drugs or as free drugs in drinking water.

No detectable bacilli were noted in the spleens of animals which received chemotherapy as free drugs after six weeks of treatment (the detection limit is 10 cfu), suggesting complete clearance. In contrast, detectable burden levels, although low, were measured in CT-NP treated mice. The relative bactericidal activity after nine weeks of chemotherapeutic treatment in the lungs and spleens of *M.tb* infected mice with free drugs administered in water resulted in clearance of bacilli in the lungs (undetectable levels <10 cfu) in 8/10 mice and complete clearance in the spleen. Pulmonary burdens in animals treated with encapsulated drugs ranged from 1 × 10^2^–1 × 10^3^ cfu whereas complete clearance in the spleen was noted for 4/9 mice.

Figure 4 represents the pulmonary pathology of infected animals over the experimental period. It is evident that infected animals which received no chemotherapy, either in the untreated control group or in the nanoparticle control group, presented with a pulmonary pathology characterised by distinct lesion formation. The extent of lesion formation was indistinguishable between the two groups. Administration of therapy either as free drugs administered in water (CT-water) or in encapsulated form (CT-NP) resulted in a distinct reduction in the number and prominence of the lesions after four weeks. Differences in the extent of lesion formation were difficult to distinguish at the 4-week time point. Pulmonary pathology was further reduced after six weeks of therapy and even more after nine weeks of treatment. On careful examination, it appeared that reduction of lesions in animals treated with free drugs was superior compared to animals treated with encapsulated drugs after six weeks, but pathology appeared largely indistinguishable after nine weeks. This study was again repeated for a 4-week treatment duration with the free drugs being administered by oral gavage instead of being available in the drinking water with a similar outcome (results not shown).

Several positive outcomes of this study were found by making use of an easily scalable spray-drying technology to produce formulations in the nano-size range, namely the experiments reproducibly demonstrated that chemotherapy by nanoencapsulated anti-TB drugs matches the effect of administration of free drugs in respect of (a) controlling the inflammation, (b) the bactericidal properties, and (c) the non-toxic nature of the particles.

## 4. Discussion

Poor patient compliance represents a major challenge for drug-susceptible and drug-resistant TB treatment. Apart from severe side effects, another factor for poor patient compliance is the daily drug regimens that are extended over at least nine months of treatment. Although the latter is managed by directly observed treatment (DOTS) protocols, it is cumbersome to both the patients and clinical staff [4,21]. Therefore, the focus of this study was to test the possibility of reducing the normal daily treatment regimens of the standard chemotherapy to a once a week dose of the easily scalable reformulated anti-TB nano-drug delivery system.

Three first-line anti-TB drugs, INH, RIF and PZA, were incorporated separately into the biodegradable PLGA particles through a unique spray-drying encapsulation technology. This resulted in free-flowing powders in the nano-size range, which could be easily dispersed into solution. Due to the varying physical properties of the drugs, their encapsulation efficiencies differed. Therefore, these compounds were encapsulated separately to rule out any bias on the dosing concentrations for the animals as compared to combined formulations where mean drug encapsulation efficiencies were used [5,6].

During the manufacturing process, these PLGA particles were coated with a PEG/chitosan solution in order to reduce protein binding and extend the circulation time of the particles in the blood [16]. Previously we reported on the in vivo drug release and distribution and indicated that such coated PLGA particles did not induce any adverse effects and could extend the release of the chemotherapeutic payload in the blood over a period of five days compared to administered free drugs that were practically all removed within 16 h. Drug distribution was also observed for up to 10 days in the liver and lungs [16,17,18]. Taking these encouraging results into account, an *M.tb* H37Rv infected mouse model was used to evaluate the efficacy of these nanoencapsulated drugs in a once-off weekly dose, compared to the conventional unencapsulated drugs that were administered daily.

Upon infection, disease manifestation was allowed for three weeks whereafter the treatments were initiated for periods of four, six, and nine weeks. Lung and spleen weights were used as indicators to reflect any inflammatory responses [22,23]. The data obtained confirmed the safe use of these nanoparticles as TB-induced inflammatory responses were effectively controlled compared to the untreated group. The anti-TB drug encapsulated nanoparticles induced a resolution of infection-related inflammation of the lungs after six weeks of treatment and after nine weeks of treatment also in the spleen, similarly to what was observed for the group that received therapy with free drugs.

The rationale behind the increased dose that was used for the once a week encapsulated drugs compared to the daily dose was due to increasing reports in the literature on the limited uptake of polymeric nanoparticles after oral delivery, most probably due to the gastrointestinal mucus barriers. One such example was when samarium oxide- encapsulated PLGA nanoparticles, after oral delivery in rats, were shown to be distributed throughout most organs but the majority of particles remained within the intestines [24].

The bactericidal efficacy of a once-off weekly dose was similar when compared to the conventional daily dose treatment. The data clearly indicated effective inhibition of bacilli replication with chemotherapy treatment delivered in the form of free drugs in drinking water. Importantly, the data showed that treatment with drugs encapsulated in PLGA and delivered once every seven days by oral gavage is equally effective to inhibit *M.tb* replication. More specifically, there is a distinct benefit for longer treatment with encapsulated particles when comparing data of for weeks to that of six and nine weeks of treatment. Treatment with drug encapsulated nanoparticles also resulted in a rapid decline in bacilli burden, demonstrating effective delivery of drugs by nanoparticles. Although the rate of mycobacterial killing using free drugs was higher compared to the encapsulated drugs for the time period, it was promising to note that there was a tendency towards bacterial clearance and that the rate of bacterial killing remained consistent after four weeks when treating animals with drug encapsulated nanoparticles.

These observations are in accordance with previous studies done on polymeric particles where the potential to reduce the dosing frequency was shown without compromising the efficacy of the drugs [6,7,25,26,27,28,29]. The most prominent difference compared to what was already reported in the literature is the easily scalable spray drying technology used with the PEG/chitosan coating in order to increase the residence time [30].

Apart from solving the shortfalls of free drug treatment, another advantage of nanoencapsulated anti-TB drugs to consider is their ease of manufacturing in terms of scalability and cost-effectiveness [23]. The new study not only shows the feasibility of reducing the anti-TB drug daily regimen to a weekly regimen, but also provides the proof of principle of an easily scalable nanoencapsulation technology. Many of the new and innovative nanomedicines published to date have limited impact due to the challenges and costs relating to the process of large-scale production from sourcing raw materials to the encapsulation technology itself.

## 5. Conclusions

Reformulation of existing anti-TB chemotherapy by nanoencapsulation holds various advantages, clinical data, stretching over several decades, of the actual anti-TB drug gives guidance to the limitations that should be addressed by reformulation into nanoparticles. These nanoparticles could offer the advantage to introduce slow-release of the drug, leading to the opportunity of reducing the number of administrations and increasing patient compliance. In addition, slow-release provides the benefit in reducing the systemic exposure and thereby the toxic side effects of the drug itself. Encapsulation also protects the pharmaceutical active from the acidic gastric environment for oral delivery and is particularly relevant for molecules such as RIF that is degraded at such low pH values [31]. A lot of potential value still holds in the reformulation of the existing anti-TB drugs, while the progress towards the development of new and novel drugs is still quite slow [3]. The nanotechnologies being developed and tested in this study have the potential to create more effective and compliant treatment outcomes by a significant reduction in the frequency of drug administration. The unique spray-drying technology is anticipated to make it fully scalable to industrial level, allowing for an easier entry into large scale production for rapid clinical trialling and uptake into the market.

## Figures and Tables

**Figure 1 nanomaterials-09-01167-f001:**
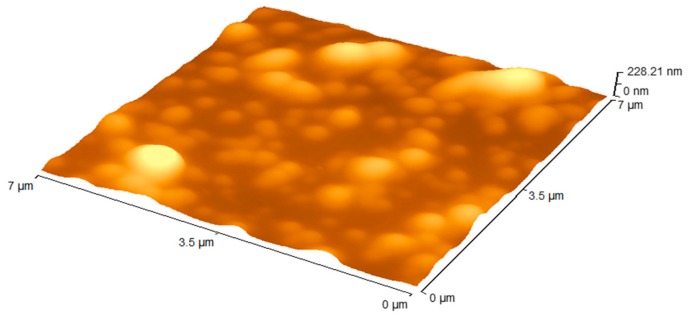
Atomic force microscopy (AFM) topography images of a representative sample of the nanoparticle formulations tested, 7 μm × 7 μm 3D rendition.

**Figure 2 nanomaterials-09-01167-f002:**
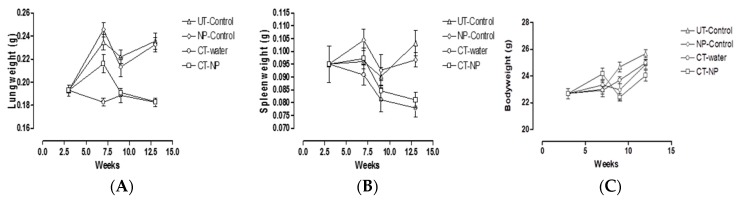
Overall kinetic changes in (**A**) lung, (**B**) spleen, (**C**) entire body, over the experimental period of 12 weeks.

**Figure 3 nanomaterials-09-01167-f003:**
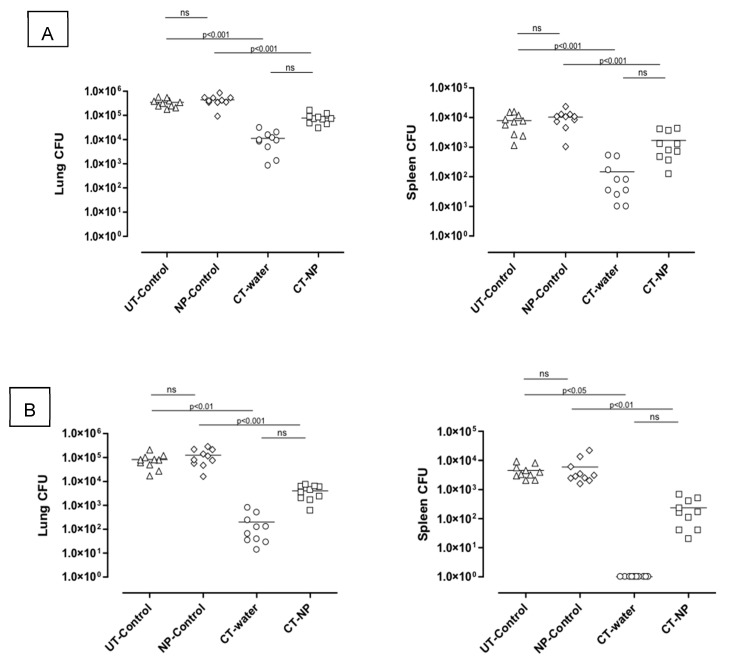
Bactericidal efficacy kinetics of treatment delivered in lungs and spleen of mice after (**A**) four, (**B**) six, and (**C**) nine weeks treatment period with encapsulated and unencapsulated INH/RIF/PZA (ns: not statistically significant).

**Figure 4 nanomaterials-09-01167-f004:**
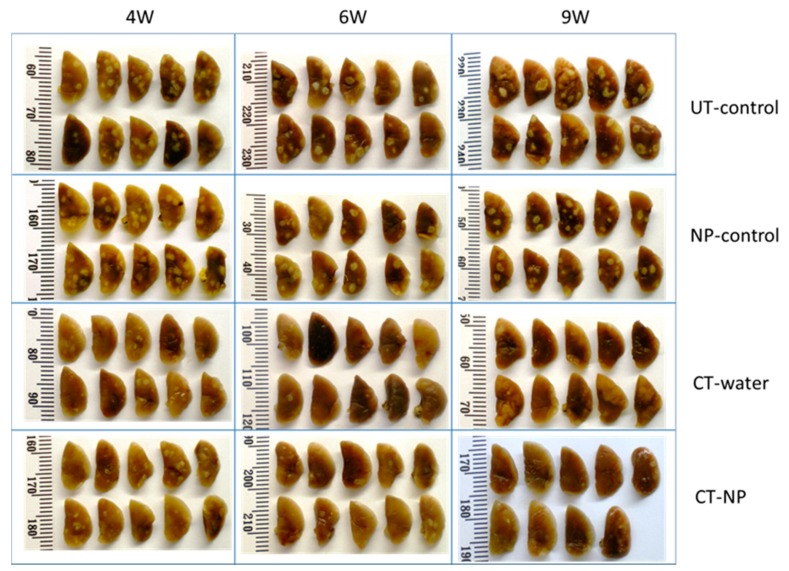
Visual representations of pulmonary pathology of infected animals over the experimental period with encapsulated and unencapsulated INH/RIF/PZA at four, six, and nine weeks of treatment.

**Table 1 nanomaterials-09-01167-t001:** Physical characteristics of nanoparticles prepared by the modified double emulsion solvent evaporation spray-drying technique.

Sample	Size (nm)	PDI *	Zeta Potential (mv)	EE ** (%)	Drug Loading (%)
PLGA-INH	328.7 ± 32.9	0.2 ± 0.01	17.7 ± 1.6	62.4	24.1
PLGA-PZA	348.3 ± 44.2	0.3 ± 0.02	19.4 ± 1.4	75.2	19.5
PLGA-RIF	252.2 ± 17.7	0.2 ± 0.01	17.9 ± 1.1	82.2	9.0
PLGA-DRUG FREE	259.6 ± 2.6	0.1 ± 0.01	11.4 ± 2.1	N/A	N/A

* PDI: Polydispersity index for size distribution, ** EE: Encapsulation efficiency (calculations as previously shown [18]).

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
