# Peer review of "Spray-Dried, Nanoencapsulated, Multi-Drug Anti-Tuberculosis Therapy Aimed at Once Weekly Administration for the Duration of Treatment"

_nanomaterials, 2019, doi:10.3390/nano9081167_

Round 1

Reviewer 1 Report

The manuscript entitled " Spray-dried, nanoencapsulated, multi-drug anti-tuberculosis therapy aimed at once weekly administration for the duration of treatment" written by Lonji Kalombo and co-authors aims to produce a scalable nanoparticulate spray-drying technology for antitubercolosis therapy able to administrate a reduced dose\time to improve treatment. Their results have demonstrated how nanoformulation of drugs can reduce  dose frequency administered and have positive effects in the treatmemt of TB. Paper is well structured and conclusions are supported by interesting data. I would suggest a minor revision after following\addressing  few issues listed below:

1) Absence of TEM\SEM or AFM images is relevant. Please provide\add  these data.

3) In vivo experiments:  lung weight comparison could be more datailed\discussed\explained.

3) Update\add references literture concerning other type of mices\NPs. Please discuss\elaborate this issue thoroughly.

4) What are the main factors influencing the way of drug-loaded NPs administration (eg oral?)?

5) What are perspectives in the application to patients problematics? Is the scale up of this technology possible? Please elaborate\discuss further.

6) A moderate English revision will further improve the comprehension of manuscript.

Author Response

Please see the attachment for additional information.

Please note the line references refer to the document in "track changes" mode

We would like to refer the reviewer to the AFM data line 141 to 145 and line 233 to 245

Additional explanations were inserted in the manuscript in line 270-304

We would like to refer the reviewer to line 52-79 in the manuscript.

We would like to refer to the inputs in line 57 to 63.

We would like to refer the reviewer to line 80 to 89 to supply information on what drug limitations / patient compliance we aim to address, as well as line 428 to 435 and line 450 to 454 in the manuscript to confirm scalability.

English revision done as seen in track changes throughout document

Reviewer 2 Report

The manuscript entitled “Spray-dried, nanoencapsulated, multi-drug anti-tuberculosis therapy aimed at once weekly administration for the duration of treatment” approaches the nanoformulation of three frontline antitubercular drugs encapsulated in PLGA nanoparticles. The encapsulation of these three drugs in PLGA nanoparticles with similar encapsulation efficiencies has been described before (Ref. 6). The difference is the encapsulation method. The authors in this manuscript used a modified double emulsion solvent evaporation spray-drying technique published previously by Semete et al. (Ref. 15). Thus, the novelty of this manuscript is the application of the spray-dried technique to a known nanoformulation, which raises issues about the novelty of this work. 

I list my specific concerns for the manuscript below. My comments are mainly aimed at tightening up the logic and clarity of what you have communicated and tested.

1.    All the main components are present in the abstract, but the encapsulation efficiencies could appear in ascending order.

2.    The Introduction is well-written and concise, and the hypothesis and purpose of the study are clearly presented but this important review (Mater Sci Eng C Mater Biol Appl. 2018 Dec 1;93:1090-1103. doi: 10.1016/j.msec.2018.09.004) regarding polymeric delivery systems for pulmonary administration of anti-TB drugsis missing. In addition, how the methodology described in this manuscript is an advanced in the field considering the several nanoformulations which already exists in the literature.

3.    In the Nanoparticle preparation and characterization description the ethyl acetate and phosphate buffered saline volumes are missing.

4.    Line 142. Please change ml to mL.

5.    I was quite disappointed with the obtained nanoparticle size. Nowadays, nanoparticles with a size below 100 nm have the highest potential, which relates to their ability to circulate in the blood for long periods of time and biodistribution patterns. But for an aerosol inhalationmust be different. The authors could insert some information about the ideal size and shape of the nanoparticles for this particular application.

6.    The particles need to be investigated by scanning electron microscopy for example. The characterization by DLS is not enough and the results should be complemented with the correlation curves.

Author Response

Please see the attachment for additional information.

Please note the line references refer to the document in "track changes" mode

We thank the reviewer for the suggestion and the correction was made in line 20 of the manuscript.

The suggested review was incorporated into the introduction section line 56-58 of the manuscript.

The suggested corrections were incorporated into the manuscript in line 125

The error was corrected in line 142 (now line 171)

This observation is very relevant for pulmonary delivery however not as important for oral drug delivery. Research has reported the ability of nanoparticles to bypass oral and intestinal absorption barriers and deliver drugs to the site of action. This usually occurs due to the smaller size (typically ranging from one to several hundred nanometers) and appropriate charge with a higher surface-volume ratio. The authors have also shown this in several in vivo studies (Semete et al 2010, 2012, Booysen 2013). The authors would like to pursue the option of inhalable therapy in future research.

We would like to refer the reviewer to the AFM data line 141 to 145 and line 233 to 245

Round 2

Reviewer 2 Report

The authors significantly improved the article. I have no further questions.